# MITO-VATION: Feasibility of a technology-supported structured home exercise program in Mitochondrial Disease

Jeremey Thomas Horne[1,2‡*], Natalie E. Allen[3☍], Serene S. Paul[3☍], Judith Walker[1☍], Carolyn Sue[1☍]

**1** Neuroscience Research Australia (NeuRA), Randwick, Australia, **2** Calvary Health Care, Kogarah, Australia, **3** Faculty of Medicine and Health, The University of Sydney, Camperdown, Australia

☍ These authors contributed equally to this work.
‡ This author led to this work.
* jeremy.horne@health.nsw.gov.au

## Abstract

Exercise intolerance, combined with low levels of physical activity, are commonly observed in individuals with Primary Mitochondrial Disease (PMD). However, access to health professionals with expertise in prescribing exercise to this population is limited. The use of digital health technology (DHT) may be a feasible and acceptable approach for therapists to support people with PMD to increase levels of physical activity, including exercise. Ten participants with mild to moderate PMD were recruited. All were provided with an eight-week home exercise program via an online exercise prescription app and remotely monitored using a smart watch. Participants received telehealth supporting their home exercise regimen along with reminders to move from the smart watch. The primary outcomes were feasibility and acceptability. Secondary outcomes were physical performance measures and fatigue, measured pre- and post-intervention. Only 26% of eligible participants enrolled. There were no dropouts, and four minor adverse events reported. Most participants (80%) participated in 80% or more of the telehealth sessions and wore the smart watch on 80% or more days during the study. Daily step target achievement was poor and only one participant met their individualised target on ≥80% of days. Half the participants achieved their weekly target of 150 intensity minutes (heart rate >50% of their theoretical maximal heart rate) on 7 or more weeks. Home exercise program adherence was low with only 30% of participants completing 80% or more of the scheduled strength training sessions over 8 weeks. Post-hoc exploration found pre-intervention exercisers achieved 4 out of 5 intervention targets compared to 0 for non-exercisers. Acceptability outcomes extracted from post-program questionnaires were overall positive towards the smart watch and home exercise program. There were no meaningful changes in any physical outcome measures or fatigue post-test. The use of DHT may be feasible and acceptable for prescribing home exercise and monitoring activity

which permits unrestricted use, distribution, and reproduction in any medium, provided the original author and source are credited.

**Data availability statement:** All relevant data are within the manuscript and its Supporting information files.

**Funding:** JH recieved a research fellowship grant from the Mito Foundation. https://www. mito.org.au/clinical-fellowships/ The mito foundation played no role in the study design, data collection and analysis, decision to publish or preparation of the manuscript.

**Competing interests:** The authors have declared that no competing interests exist.

levels in individuals with mild to moderate forms of PMD, particularly those with a history of exercise.

## Author summary

This study looked at whether using digital health technology can help adults with primary mitochondrial disease (PMD) take part in a safe, structured home exercise program. PMD is a rare, lifelong condition that causes fatigue, muscle weakness and exercise intolerance, making it hard for people to be active and to access specialist physiotherapy, especially outside major cities. We worked with 10 adults with mild to moderate PMD who could attend our Sydney centre. Each person completed an 8-week home program that included daily step goals, weekly "intensity minutes", and strength exercises prescribed by a physiotherapist. Participants wore a Garmin Vivoactive 5 smart watch, used an exercise app (PhysiApp), and had weekly telehealth catch-ups for support and monitoring. We found that once enrolled, participants were willing and able to complete the program: no one dropped out, and there were only minor, manageable side effects such as temporary muscle soreness. Most people wore the watch as requested and valued the feedback and reminders it provided. However, keeping up with the more effortful parts of the program (especially strength training) was challenging over time, particularly for people who were not already exercising. Overall, remotely monitored exercise using consumer technologies appears safe, acceptable and potentially scalable for people with PMD, but programs must be flexible, individually tailored and supported over the longer term.

## Introduction

Primary Mitochondrial Disease (PMD) is an umbrella term used for a heterogenous group of rare genetic disorders arising from mutations in nuclear or mitochondrial DNA resulting in impaired energy metabolism. It is the most common neuromuscular defect for which there is currently no known cure and only symptom management [1]. Mitochondrial dysfunction can affect any organ or tissue throughout the body. PMD is associated with a broad spectrum of clinical presentations [2]; everyone has their own unique clinical presentation due to genetics, lifestyle, ageing and the environment in which they live. The global prevalence of PMD is estimated to be approximately 1 in 5,000 individuals, although rates vary between populations due to diagnostic and genetic differences [3]. In Australia, it is estimated that at least 1 in every 200 people carry a disease-causing mitochondrial DNA mutation [4]. The estimated average annual cost of PMD was AU $112,721 per household [5].

Exercise (a type of physical activity that is planned and structured and aimed at improving or maintaining physical fitness or overall health) [6] remains the only known physiological stimulus of mitochondrial biogenesis [7,8]. Several studies have

shown that exercise therapy can help through enhanced mitochondrial function and therefore may support a more active lifestyle [7–13]. When energy levels are low, people often lack the motivation to participate in exercise, particularly if they have limited knowledge about exercise and confidence in their ability to perform it. Consequently, people with PMD rarely achieve the recommended levels of physical activity (i.e., 150 min of moderate to vigorous exercise per week [14]). Exercise intolerance, muscle weakness, fatigue and a resulting sedentary lifestyle are the hallmarks of PMD [15].

The scarcity of clinics for PMD contributes to a shortage of health professionals, including physiotherapists, with the necessary expertise to effectively manage and facilitate exercise for people with PMD. This challenge is exacerbated in rural areas, where access to health professionals, especially those experienced in rare conditions such as PMD is particularly limited. One way to address the dual problems of low access to health professionals and low physical activity is through remotely monitored digital health technology (DHT). Currently, limited evidence exists on the use of DHT (e.g., smart watches, phones and exercise apps) to compliment a home exercise program for increasing physical activity in this population.

Subjective measures of physical activity can be unreliable [16], while objective measurements using technology (e.g., via a smart watch) is gaining acceptance [17,18]. Additionally, smart watches enable ongoing self and remote monitoring as well as programmed messaging which can increase motivation and exercise adherence [19]. Exercise apps which include narrated exercise videos, encouraging emails and the ability to videoconference are another valuable tool to help people with PMD to stay on track with their exercise program. Developing a structured exercise program that is feasible and acceptable for people with PMD is more likely to succeed with educational, behavioural and environmental support using DHT. The implementation of DHT to improve adherence to a customised exercise program is scalable and inexpensive. If shown to be feasible and acceptable, this intervention has the potential to positively impact the lives of individuals diagnosed with PMD and the wider PMD community by increasing access to specialist allied health professionals across Australia.

The aim of this study was to evaluate the feasibility (recruitment, retention, adverse events, adherence) and acceptance of DHT (Garmin Vivoactive 5 smart watch, PhysiTrack exercise program platform and telehealth) in delivering and monitoring a remote, structured home exercise program for people with PMD. A secondary aim was to quantify any changes in walking speed and endurance, muscle strength, balance and fatigue, after the 8-week remotely monitored home exercise program.

## Methods

### Ethics statement

This study was approved by the South Eastern Sydney Local Health District Human Research Ethics Committee (HREC 2024/ETH00250). Written informed consent was obtained from all participants prior to data collection.

### Study design

This study was an 8-week home-based, pre-post single group feasibility study. Participants underwent pre and post intervention assessments in the week before and after the intervention, including wearing the smart watch for one week prior to intervention to record baseline measures. Participants were then prescribed a web-based home exercise program administered by a physiotherapist and provided a smart watch to monitor activity. The study was conducted through the Australian Mitochondrial Disease Centre at Neuroscience Research Australia (NeuRA).

### Participants and recruitment

Participants with a confirmed or highly suspected diagnosis of PMD were recruited. To be eligible participants had to be ≥ 18 years of age, able to walk unaided a minimum of 6 minutes with or without rest and be willing and able to comply with the study procedures. Volunteers were excluded if they had atrial fibrillation or untreated symptomatic cardiac

arrhythmia, were non-English speaking, were pregnant or trying to fall pregnant, were wheelchair dependent, had visual acuity less than 6/60 (Snellen Test) or severe co-morbidities (including osteoarthritis, chronic obstructive pulmonary disease, cognitive impairment, depression) that would impact participant adherence, safety or interpretation of results. Participants all owned a smart phone, had home internet access and were prepared to download 2 apps (Garmin Connect and PhysiApp) to use for the duration of the study.

**Intervention**

All participants were prescribed by a single physiotherapist a structured home exercise program which was designed to be practical, safe, required minimal equipment and space to be performed competently. Participants were provided with a smart watch (Garmin Vivoactive 5) to wear on their non-dominant wrist throughout the duration of the study. This was an upgraded version of the Vivoactive 4, which offers validated step-count accuracy, long battery life, and user-friendly design [20,21]. Its heart rate and variability measures closely align with ECG standards, providing reliable data for research use [22]. Participants were directed to wear the smart watch 24 hours a day, 7 days a week (excluding showering and device recharging). Days with ≥4 hours of no data during waking hours were considered non-wear days and these data were excluded from analysis. A 10-minute training session was held at the end of the pre-intervention assessment session to teach the participants how to wear, charge, and operate the smart watch. Participants were also given a demonstration on how to use the Garmin Connect app and were encouraged to self-monitor their daily activity levels and work toward meeting their daily and weekly targets (as described below). Hard copy instructions regarding the home exercise program, smart watch, Garmin Connect and PhysiApp were supplied to all participants.

The home exercise program was prescribed using the web-based programming application PhysiTrack and accompanying end-user phone app, PhysiApp. Prior studies have demonstrated that PhysiTrack improves exercise adherence and engagement compared with paper-based programs and has high usability and patient satisfaction across neurological and musculoskeletal populations [23–25]. The PhysiTrack application was used by the physiotherapist to provide instructions regarding the home exercise intervention while PhysiApp allowed participants to view their exercise program via phone or tablet. The home exercise intervention lasted 8 weeks and consisted of the following five components: 1) A daily step target. 2) A weekly intensity minutes target. 3) Major muscle group strengthening exercise 3 times per week. 4) Weekly catch-up sessions. 5) Watch wear targets. No medication or dietary modifications were made during the trial.

1)  Daily step target

Participants agreed to a daily step target based on their 7-day pre-intervention baseline smart watch data. The daily step average over this week was used as the starting target for week 1 of the intervention. This target was increased by a maximum of 10% per week if the target was met on ≥4 days in the preceding week. The total increase over the entire intervention would not exceed 50% of the baseline measure. The means by which a daily step target was achieved remained at the discretion of each participant (i.e., planned walk or increased habitual activity). Participants were encouraged to regularly view their number of steps taken to enhance motivation. If the daily step target was achieved, the watch would vibrate and provide congratulatory messages.

2)  Intensity minutes target

A Garmin proprietary algorithm assessed activity intensity and labelled the data as 'intensity minutes', which were earned based on a participant's heart rate relative to their average resting heart rate. Intensity minutes were earned when the participant's heart rate exceeded 50% of their theoretical maximum, predicted by the Garmin default equation (220 – age), and sustained at this level for at least 10 minutes. If a participant's heart rate increased above 70%, they were rewarded with 2 minutes for every one minute performed. Participants were encouraged to achieve ≥150 intensity minutes each week by maintaining their heart rate within 50–70% of their predicted maximum during exercise bouts of ≥10 minutes.

3) Strength training target

Each participant was provided with an individualised home strength training program aimed at improving strength and range of motion. The program utilised elastic resistance bands of varying resistance (1.3 kg, 1.7 kg, 2.1 kg and 2.6 kg) to perform 3 times per week on non-consecutive days. The home exercise program included a short warm up (5-min walk) followed by 6 stretching exercises targeting the lower limb, upper limb and spine. Once warmed up, participants performed a circuit of 8 multi-joint strength exercises involving large muscle groups (lunges, row, sit to stand, chest press, hip abduction, trunk rotations, hip extensions and heel raises). Participants performed 1–3 rounds of single sets (10–15 repetitions) to allow full muscle group recovery between sets. The PhysiApp included fully narrated exercise videos and exercise information ensuring participants had a clear understanding of their exercise program and targets each week. A hard-copy version was also supplied as a part of the study exercise diary (S1 Fig).

Immediately following the baseline assessment but prior to the start of the intervention period, participants were oriented to their exercise program. A hard copy exercise program was supplied to participants including an exercise diary to be completed after each session to record the date, sets, repetitions, resistance band level and rating of perceived exertion for each individual activity. Strength exercises were performed in a specific order (i.e., alternating arm and leg exercises to minimise muscle soreness and fatigue). Sets and repetitions for each participant, along with training intensity (i.e., elastic band resistance) were determined using pre-intervention assessment baseline data. The modified Borg rating of perceived exertion scale (1–10) was used with a score of 6 (i.e., able to speak 3–5 words comfortably) being the starting level. Once a score of 5 (i.e., able to speak more than 5 words comfortably) or less was perceived, the band resistance was increased to the next level. The speed of repetitions focused on having a controlled concentric/eccentric movement phase. Training volume (sets x reps x band resistance) was increased slowly as tolerated. Volume targets were emailed to participants weekly through PhysiTrack. Targets were discussed at the weekly telehealth catch-up session. Participants completed the supplied exercise diary (S1 Fig) for monitoring adherence to the prescribed exercise sessions as well as documenting any barriers or adverse events encountered.

4) Weekly catch-up session target

Weekly telehealth catch-up sessions took place at the beginning of each intervention week via phone, video call (using PhysiTrack) or email, based on participant preference. Each session lasted 10–15 minutes and aimed to monitor adherence, identify adverse events and address any barriers to exercise engagement. Prior to each session, the study physiotherapist remotely reviewed participant data via authorised access to participants' Garmin Connect account, including metrics such as daily steps, heart rate, intensity minutes, activity and sleep patterns. Key data were transferred to secure participant files at NeuRA. This process enabled the physiotherapist to remotely monitor responses to the exercise program and use the information to tailor exercise prescription, discuss progress and address participant queries.

5) Watch wear target

Participants were asked to wear the smart watch continuously over the 8 weeks of intervention. They downloaded the Garmin Connect app onto their smart phone and transferred data regularly to avoid any potential loss of information.

**Outcomes**

The primary outcome was feasibility, which was measured via recruitment, retention, adherence and adverse events to the exercise prescription. *Recruitment* was the percentage of participants who enrolled in the study from the eligible pool, and feasibility was a priori set at 50%. The *retention* rate was the percentage of participants who completed the post intervention assessment out of the total number of enrolled participants, with feasibility set at 80% retention. Exercise program *adherence* (% prescribed exercise sessions completed) was set at ≥80% for each of the five individual components (see Table 2 for details). *Adverse events* were defined as any intervention-related event resulting in an inability to perform

exercise or alteration to the home exercise program and were reported to the study physiotherapist during the weekly telehealth catch up session.

Secondary outcomes were walking speed, walking endurance, lower limb muscle strength, upper limb muscle strength, balance and fatigue, each measured pre and post intervention. Walking speed [26] was measured in metres per second using a 6m long electronic (Zeno) walking mat [27] with the mat placed in the middle of a 10m walkway. All participants performed 4 passes on the mat walking as fast as possible with the average speed recorded. Walking endurance was measured as the distance walked in a 6-minute walk test, conducted in a 20m corridor [28]. The 5 times sit to stand test was performed using a standard height chair (43–45 cm) and stopwatch to record the time (s) taken to stand up and sit down five times [29]. Grip strength (kg) in sitting was measured using a Jamar handheld dynamometer where a maximal isometric contraction was held for 3–5 seconds [30]. A total of 3 trials were performed with the best score being recorded for each hand, noting hand dominance. Balance was assessed with the Berg Balance Test, a 14-item scale designed to measure balance [31]. A score out of 56 was recorded with a higher score indicating better balance. Fatigue was assessed with the Fatigue Impact Scale (FIS), a self-reported questionnaire evaluating the effect of fatigue on three domains of daily life: cognitive functioning, physical functioning, and psychosocial functioning [32].

At the post intervention assessment session, participants completed bespoke questionnaires to evaluate their experiences and perceptions of the smart watch, the home exercise program and PhysiApp platform. Participants rated the usability, helpfulness and desirability through a mix of Yes/No, Likert scale and open-ended questions. Participants were asked if they found the DHT easy to use, what they liked or disliked, how they used it, and if they would continue to use it if it were made available long term.

### Statistical analysis

This pilot study aimed to determine the program's feasibility and acceptability in people with PMD, and no formal sample size calculations were performed [33]. The primary outcome (feasibility) was calculated from a combination of the percentage of participants who enrolled in the study from the eligible pool, percentage retained, percentage who adhered to the exercise intervention (defined as completing ≥80% of the prescription), and the number and severity of adverse events. We conducted an exploratory post-hoc analysis to explore if exercise history influenced adherence to the program. Acceptability outcomes were extracted from the post-program questionnaires. For secondary outcomes, descriptive statistics of outcome measures (means and standard deviations for continuous outcomes, number and percentage for categorical data) were generated to visually inspect any changes from pre to post intervention.

### Results

Of the 39 clients attending the Australian Mitochondrial Disease Centre in person between July and November 2024, 10 were included and completed the study (see Fig 1).

Participant characteristics at baseline are shown in Table 1 (see S1 Table medication list and S2 Table participant symptoms for more group data). There were 10 participants (age range 29–76 years, 3 (30%) male) who had mild to moderate PMD. Only three (30%) participants reported undertaking regular exercise prior to the study.

### Feasibility outcomes

Recruitment reached 26% (10 of 39 eligible participants enrolled), which was below the 50% target (Fig 1). This was largely due to geographical barriers, as most eligible participants resided outside metropolitan Sydney and could not attend the two in-person study appointments. Contrastingly, retention was 100% suggesting the intervention using DHT was feasible for those who could attend. Minor adverse events were reported by four participants, including muscle soreness (one participant) and exacerbation of pre-existing joint pain (three participants). All cases were managed with minor

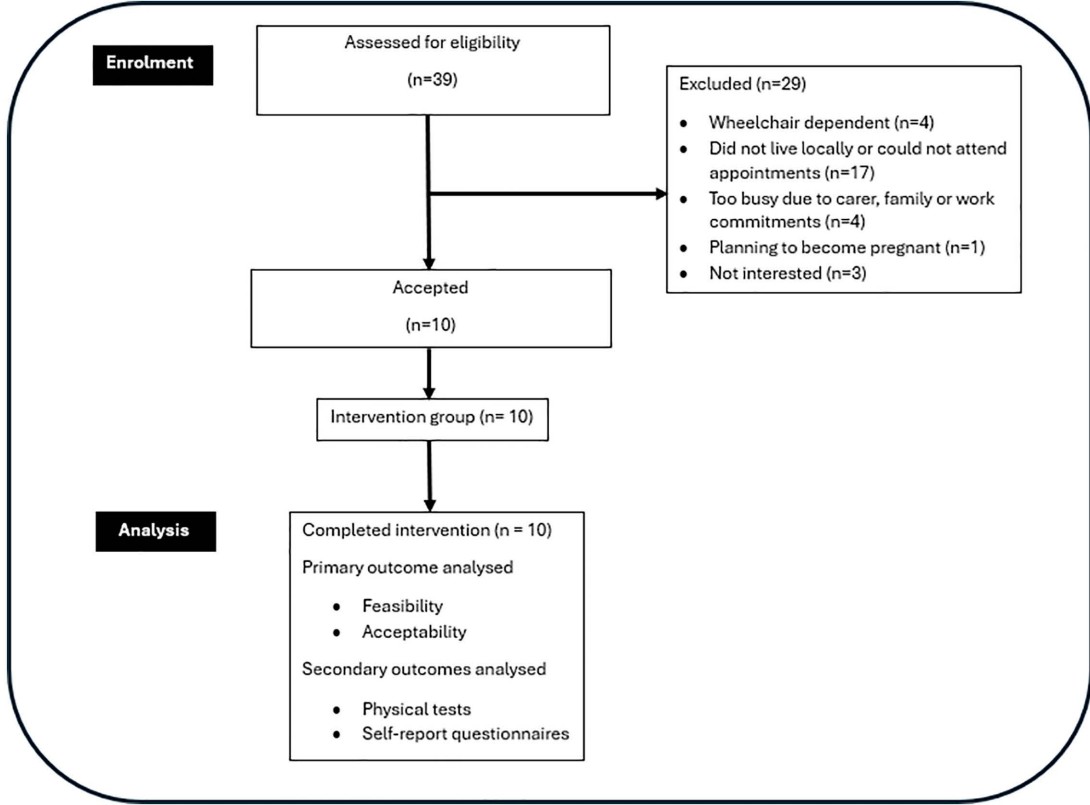

**Fig 1. Flowchart of participants through the study.**

program adjustments. No serious adverse events occurred, supporting the safety of remotely monitored exercise in this population. Adherence to the five intervention targets varied (Table 2) reflecting individual differences in engagement and capacity.

Only one participant achieved the daily step target on ≥80% of days (Table 2). Nevertheless, 90% of participants increased their overall daily step count average between baseline and post intervention (Fig 2). The average number of steps taken per day by the group over the 8-week intervention were 6357 (±1755).

Half the participants achieved the weekly intensity minutes' target of ≥150 minutes of moderate to vigorous intensity exercise on at least 7 of the 8 intervention weeks (Table 2). Average weekly intensity minutes ranged widely, from 36 to 2637 minutes per participant (Fig 3). The two participants with the highest intensity minutes (>2000 min/week) had Type 2 diabetes and elevated resting heart rates, likely contributing to the accumulation of intensity minutes during routine daily activities rather than structured exercise. Overall, 74% of recorded intensity minutes were at moderate intensity (50–70% of maximum heart rate) with 24% at vigorous intensity (>70% of maximum heart rate).

Adherence to the strength training target of ≥19/24 sessions was achieved in only 30% of participants. Session completion percentage across all ten participants for the first week of the intervention was 87% but dropped to 23% by the final week (Fig 4), with a continual decline occurring after week 3.

Telehealth catch-up session attendance was high, with 80% of participants achieving the target of attending at least 7 of 8 sessions. All but one of the telehealth catch-up sessions were conducted over the phone, reflecting a strong preference for this mode over video consults. Participants cited greater privacy, comfort and flexibility, particularly for those

**Table 1. Baseline participant characteristics.**

| Characteristic | N (%) or mean (± SD) |
|---|---|
| Age (years) | 54 (±16.8) |
| Age range | |
| 20–39 years | 4 (40%) |
| 40–59 years | 2 (20%) |
| 60–79 years | 4 (40%) |
| Sex | |
| *Female* | 7 (70%) |
| *Male* | 3 (30%) |
| NMDAS; mean | 24.8 (±13.0) |
| Clinical presentation | |
| *CPEO + mitochondrial myopathy* | 5 (50%) |
| *MERRF* | 1 (10%) |
| *MIDD* | 1 (10%) |
| *Mitochondrial cardiomyopathy* | 3 (30%) |
| Race n | |
| *Caucasian* | 9 (90%) |
| *Indo-Fijian* | 1 (10%) |
| Employment | |
| *Full-time* | 1 (10%) |
| *Part-time* | 2 (20%) |
| *Disability pension* | 2 (20%) |
| *Carer* | 1 (10%) |
| *Retired* | 4 (40%) |
| Body mass index (BMI) | 27 (±3.8) |
| Number of co-morbidities | 9 (±3.4) |
| Number of medications | 6 (±2.5) |
| Exercise history | |
| *Yes* | 3 (30%) |
| *No* | 7 (70%) |
| Smart watch owner | |
| *Yes[†]* | 6 (60%) |
| *No[‡]* | 4 (40%) |

NMDAS = Newcastle Mitochondrial Disease Adult Scale, CPEO = Chronic Progressive External Ophthalmoplegia, MERRF = Myoclonic Epilepsy with Ragged Red Fibres, MIDD = Maternally Inherited Diabetes and Deafness.

[†] One owned a Fitbit watch, three an Apple watch, one a Google watch and one a Garmin watch.

[‡] Two owned an analogue watch and two had no watch.

working, as key advantages of phone calls. Two participants were unable to engage in phone or video calls due to work commitments so opted for email correspondence. However, delayed and inconsistent responses meant these participants were unable to meet the telehealth catch-up session requirements. Smart watch wear adherence was similarly strong, with 80% of participants meeting the target of ≥45 days over the 8-week period (Table 2).

A post-hoc exploratory analysis (Table 2) found pre-intervention exercisers collectively achieved 4 out of 5 intervention targets (all except daily step count), whereas non-exercisers did not achieve any of the targets.

**Table 2. Adherence to intervention targets of step count, intensity minutes, strength sessions, weekly telehealth catch up and watch wear.**

| Participant | Daily step target achieved (no. of days) | Weekly intensity minutes target achieved (no. of weeks) | Strength sessions completed (no. of sessions) | Telehealth sessions completed (no. of sessions) | Watch wear (no. of days) |
|---|---|---|---|---|---|
| Total number of sessions | *56 days* | *8 weeks* | *24 sessions* | *8 sessions* | *56 days* |
| Adherence target | *≥45 days* | *≥7 weeks* | *≥19 sessions* | *≥7 sessions* | *≥45 days* |
| 1 | 10 | 4 | 15 | **7** | **52** |
| 2 | 15 | **8** | **20** | **8** | 47 |
| 3 | 19 | **7** | 17 | **7** | **52** |
| 4 | 21 | **7** | 17 | **7** | **54** |
| 5 | 20 | 0 | **19** | **8** | **49** |
| 6 | **54** | **8** | **24** | **8** | **56** |
| 7 | 27 | 1 | 9 | 0 | 38 |
| 8 | 25 | 1 | 4 | 0 | **45** |
| 9 | 13 | **8** | 15 | **7** | 25 |
| 10 | 23 | 0 | 5 | **7** | **52** |
| Exercisers (n = 3) | 31 | **7** | **19** | **7** | **54** |
| Non-exercisers (n = 7) | 19 | 3 | 12 | 5 | 44 |

*Bold type = goal met.*

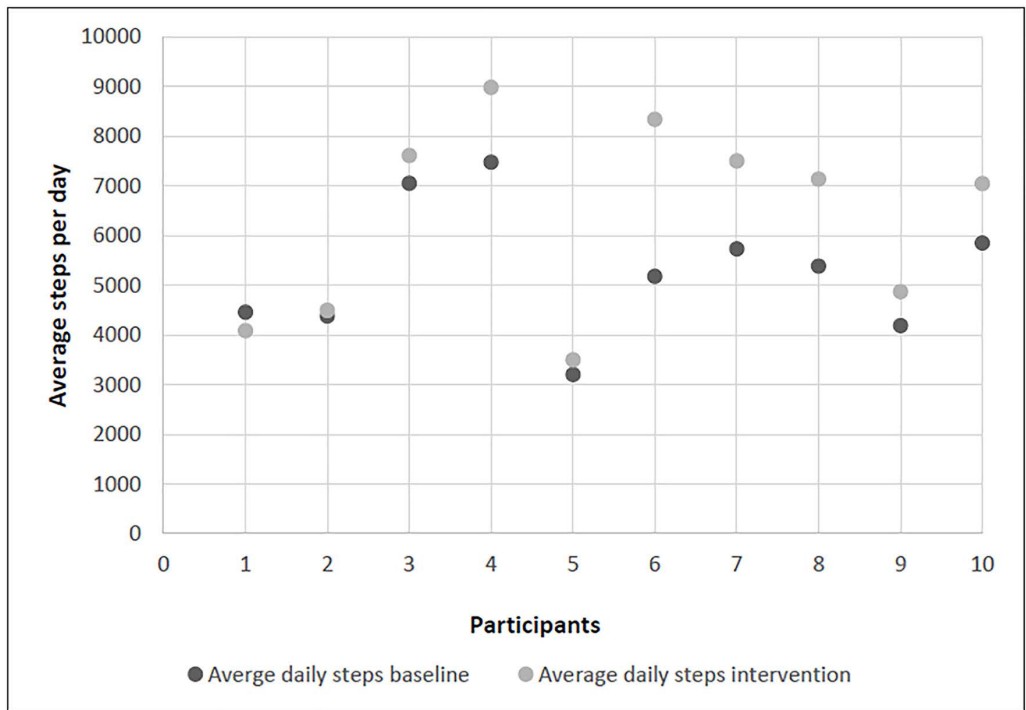

**Fig 2. Average steps per day, baseline and 8-week intervention period, for individual participants.**

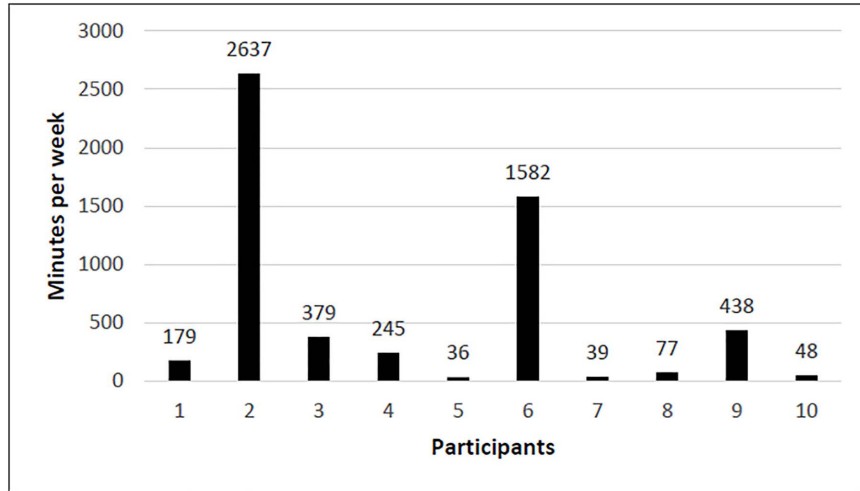

**Fig 3. Average weekly intensity minutes per participant.**

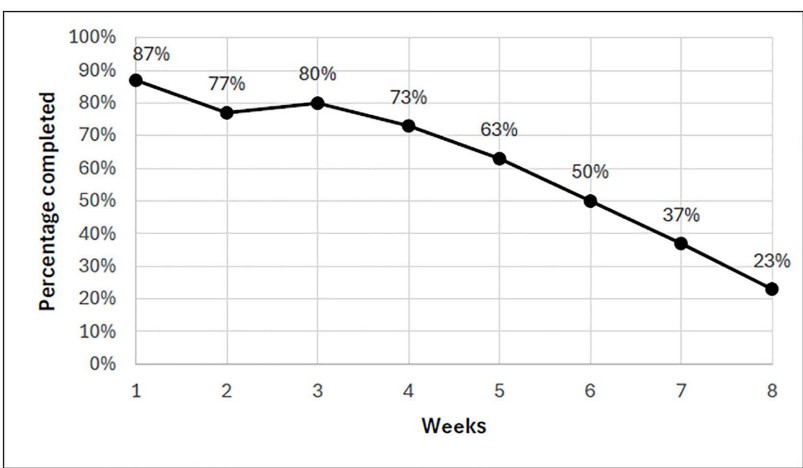

**Fig 4. Strength session completion rate per week across all 10 participants.**

## Acceptability of the intervention

The smart watch was rated as easy or very easy to use by 80% of participants, with heart rate monitoring (100%), step counting (90%) and sleep tracking (80%) being the most utilised features (S3 Table). Activity reminders were motivational for 70%, though 30% found them stressful or demotivating. While most participants (80%) reported no issues with continuous wear, 20% experienced discomfort or minor skin reactions. Notably, 70% used the watch to achieve their personal targets, and 80% intended to continue using a smart watch post intervention. Participant feedback revealed both positive and negative emotional responses to the smart watch's reminder features. For some, the reminders were motivational, with comments such as: *"[it] made you get up"* (ID 1), *"it was a reminder"* (ID 2 & 4), *"made me aware of how I was going and I liked the step count and sleep information the most"* (ID 10). In contrast, others commented the reminders were discouraging, stating they were *"very negative… the constant reminder of how badly you have done during the day... It made me lose motivation"* (ID 8).

Most (90%) participants reported that the home exercise program was easy to follow, saying it *"[was] a full body work-out"(ID 3)*, *"made me feel better when did just ten reps"(ID 5)*, *"built routine"(ID 6)* and noting they learnt *"some new exercises"(ID 10)* (S4 Table). However, 40% found the program tedious after a few weeks, and 30% reported that 3 sessions per week were too much. Reported reasons for missed or incomplete strength sessions included *"fatigue, feeling lazy/low motivation" (ID 1)*, *"not feeling well, muscles stiffness, pain" (ID 2)*, *"social events" (ID 7)*, *"carer duties" (ID 4 & 10)*, *"poor sleep and stress" (ID 4) and "holidays" (ID 7 & 8)* (S5 Table).

All participants reported personal motivation was a key factor in maintaining engagement, while catch-up sessions and the PhysiApp were less important. Ninety percent reported that perceived health improvements supported their continued participation, and 60% indicated they would continue using the exercise equipment once the study had finished (S4 Table).

Most participants (80%) used the PhysiApp program (S6 Table). Sixty percent found it easy to use, 20% were neutral, and all accessed it between one and three times per week. The video demonstrations were consistently rated as the most helpful feature. Five (50%) participants reported improved fitness and would recommend the app to others. Two (20%) participants preferred the hard copy exercise diary and used weekly catch-ups to receive updated exercise prescriptions.

### Secondary outcomes

There were no apparent group-wide changes in physical outcomes or fatigue post-intervention (Table 3).

### Discussion

This study is the first to evaluate the feasibility and acceptability of remotely monitoring a home-based exercise program for individuals with mild to moderate PMD using DHT. While recruitment fell short of the 50% target, primarily due to geographic barriers limiting access to in-person assessments, several feasibility indicators were met. There were no dropouts and only minor, manageable adverse events, supporting the program's safety in this population. Adherence to passive components, including smart watch wear and telehealth catch-up sessions, was high and participant feedback reflected positive perceptions of the program's overall acceptability.

In contrast, adherence to active components (i.e., daily steps, intensity minutes, and strength sessions) was variable with a noticeable decline in strength session adherence after week 3. Participants with prior exercise habits demonstrated stronger engagement, suggesting that prior experience with exercise may influence adherence to remote interventions supported by DHT. Overall, many feasibility metrics were met and the remaining challenges, such as exercise monotony, session frequency and recruitment limitations offer actionable insights for future studies. These barriers are modifiable through tailored and individualised program design, supporting the potential to provide safe and acceptable exercise interventions for people with mild to moderate PMD.

Recruitment (26%) was affected by geographical constraints, as most PMD clients at the Sydney-based Australian Mitochondrial Disease Centre reside outside metropolitan Sydney and could not attend multiple study visits. Additionally, reluctance to commence exercise was common among non-exercisers. These findings underline accessibility and behaviour-change barriers typical in rare diseases within Australia, where specialist clinics are confined to major health precincts. A coordinated national network of physiotherapists and exercise physiologists, linked via DHT and telehealth, could improve care delivery. Similar integrated models exist in the United Kingdom, where the NHS supports multidisciplinary mitochondrial disease centres in Newcastle, London, and Oxford [34]. Development of Australian PMD exercise guidelines, modelled on UK standards but incorporating DHT, would enhance clinical and educational outcomes.

Even though most participants failed to meet their daily step targets, 90% improved their average daily steps by 20% by study end. The rigid daily goal structure likely conflicted with the fluctuating fatigue and energy levels experienced in PMD. A more flexible approach such as weekly average step goals or 3–4 active days per week may better align with patients' variable capabilities. Evidence suggests that intermittent achievement of age-recommended step targets still offers

**Table 3. Mean (SD) pre and post intervention outcome results.**

| Outcome measure | Pre intervention (n = 10) | Post intervention (n = 10) |
|---|---|---|
| 10m Fast Walk (m/s) | 1.7 (± 0.2) | 1.7 (± 0.2) |
| 6 Minute Walk Test (m) | 474 (± 87.0) | 482 (± 71.3) |
| 5 Times Sit to Stand (sec)* | 12.3 (± 4.4) | 12.3 (± 6.0) |
| Berg Balance Test (0–56) | 54 (± 2.1) | 55 (±1.0) |
| Grip strength (kg) – dominant hand | 25.9 (± 10.2) | 27.4(± 9.4) |
| Grip strength (kg) – non-dominant hand | 24.8(± 9.7) | 25.9 (± 9.3) |
| Fatigue Impact Scale (0–160)* | 63 (± 33.7) | 66 (± 24.1) |

* higher score = worse result.

substantial health benefits and reduced mortality risk [35,36]. Adjusting frequency expectations would allow recovery while maintaining motivation and physiological benefit.

Heart-rate monitoring proved partly effective, with participants exercising between 50–70% of their theoretical heart rate maximum 76% of the time, in line with Australian care standards [37]. The commonly used maximal heart rate equation of 220 minus age [38] is known to have limited predictive accuracy [39–44] and may be unsuitable for PMD due to cardiac abnormalities [45]. The variability in weekly intensity minutes (Fig 3) illustrates this limitation, particularly among participants with diabetes, where autonomic dysfunction and elevated resting heart rate artificially inflated intensity readings [46]. Therefore, heart rate based training metrics may not accurately reflect true workload in PMD. Clinicians may need to rely on alternative intensity measures such as perceived exertion scales [47], talk tests [48], or disease-specific heart rate equations [49]. Manual activity recording or accelerometry-based analysis could further improve accuracy by distinguishing planned exercise from incidental activity.

Adherence to strength training declined to 23% by week 8, with clear evidence that week 3 marks a pivotal drop-off point. Participants attributed this to monotony, fatigue, and over-prescription (three sessions per week) alongside motivational and lifestyle barriers (S5 Table). This suggests programs for PMD should be adjusted at least every 2–3 weeks, more frequently than in general or other clinical populations [50], to sustain engagement and prevent fatigue-related withdrawal. Post hoc analysis revealed stark differences between exercisers (achieved 4/5 goals) and non-exercisers (0/5 goals), demonstrating feasibility for the former but a need for tailored introductory interventions for the latter.

Several adherence-enhancing strategies warrant exploration in future research in PMD. Exercise snacking involves incorporating short (<1 min) vigorous bouts dispersed throughout the day, which has shown high adherence (87–98%) in chronic disease populations [51–54]. Other strategies include regular exercise variation, modifying exercise content every 2–3 weeks to maintain interest and engagement. The use of hybrid care models combining remote monitoring with periodic in-person sessions to provide feedback, support, and variety [55,56] may also be beneficial. Finally, tailored onboarding for non-exercisers, starting with 1–2 sessions weekly, focusing on motivation, behavioural reinforcement, and gradual progression may further support and encourage consistent exercise adherence.

Weekly telehealth catch-up sessions were a well-accepted feature, providing reassurance, feedback, and motivation. Phone communication was overwhelmingly preferred (98%) over video and emails, mainly for convenience, privacy, and reduced technological burden. While telehealth video has benefits for visual feedback, all modes hold value in a hybrid model of care [57,58]. Two participants utilised personal exercise physiologists for occasional supervision, which supported motivation but did not improve unsupervised adherence, suggesting hybrid support models must be structured and consistent.

Smart-watch adherence was high (80%), showing strong engagement with passive monitoring. Continuous physiological tracking bridged data gaps between clinic reviews, enabling real-time assessment of fatigue, safety, and progress. Yet, device data must be interpreted cautiously as only ~4% of consumer wearables are formally validated [59], with heart rate errors of ±3% and step errors of ±10% and external factors such as fit, motion intensity, comorbidities, and medication potentially affecting accuracy [60]. Furthermore, negative feedback (e.g., low step count) may demotivate users, as reported by one participant, echoing evidence linking misleading feedback to reduced motivation and elevated stress markers [61]. Customisable notification settings may mitigate such issues.

No significant changes were detected in physical outcomes or fatigue. However, these results reflect low exercise dose due to poor compliance within a relatively short 8-week study duration, and small sample. Importantly, 80% of participants reported positive experiences and interest in continuing smart-watch use, confirming DHT's acceptability. Still, some preferred paper-based materials, highlighting the need for technology-flexible delivery options. Future large-scale randomised trials are required to validate these findings, refine DHT-based interventions, and inform Australian PMD exercise guidelines incorporating hybrid and telehealth models.

### Limitations and future directions

This pilot study has several limitations. The small sample (n = 10), restricted to individuals with mild to moderate disease, limits generalisability to people with severe PMD. Participant variability, including a wide age range and diverse clinical presentations, could not be adequately explored, though future research should examine whether age influences engagement and outcomes in DHT-supported exercise interventions. The absence of a control group, lack of assessor blinding, and short intervention duration further constrain interpretation. Reliance on self-report and variable smart-watch accuracy may have introduced bias, and levels of digital literacy and internet access may have affected participation. Declining adherence also reflected the limitations of an exercise program with limited tailoring, which does not align with best practice in clinical care. Future studies should include larger samples, longer interventions and tailored programming to better support individual needs and explore factors such as age, disease severity, and technology literacy and accessibility.

### Conclusion

This study demonstrates that remote exercise monitoring via DHT in people with mild to moderate PMD is safe, acceptable, and potentially feasible with modification. Feasibility was influenced by participants' prior exercise experience, with the current protocol working well for those already active but requiring adaptation for sedentary individuals. A sustainable, scalable approach integrating DHT within a hybrid model combining remote monitoring with regular in-person sessions warrants investigation. Expanding access to multidisciplinary clinics and ensuring frequent program adjustments (every 2–3 weeks) may improve motivation and adherence. With refinement, DHT-based exercise programs may play a key role in extending specialist physiotherapy services to individuals with rare diseases who otherwise face barriers to ongoing, accessible care.

### Supporting information

**S1 Fig. Exercise diary.**
(TIFF)

**S1 Table. Participant medications list.**
(TIFF)

**S2 Table. Participant symptoms.**
(TIFF)

**S3 Table. Summary of smart watch acceptability answers.**
(TIFF)

**S4 Table. Summary of home strength exercise program acceptability answers.**
(TIFF)

**S5 Table. Reasons for not completing a strength session.**
(TIFF)

**S6 Table. Summary of PhysiApp platform acceptability answers.**
(TIFF)

## Acknowledgments

We would like to thank the Mito Foundation for providing the necessary funding to conduct this study. We also thank Dr Karen Crawley and Emeritus Professor Colleen Canning for their insightful discussion, assistance and advice with manuscript review. Finally, we thank all the participants with PMD who gave their time and effort during this project.

## Author contributions

**Conceptualization:** Jeremey Thomas Horne, Natalie E. Allen, Serene S. Paul, Judith Walker, Carolyn Sue.

**Data curation:** Jeremey Thomas Horne.

**Formal analysis:** Jeremey Thomas Horne, Natalie E. Allen, Judith Walker, Carolyn Sue.

**Funding acquisition:** Jeremey Thomas Horne, Natalie E. Allen.

**Investigation:** Jeremey Thomas Horne, Carolyn Sue.

**Methodology:** Jeremey Thomas Horne, Serene S. Paul, Judith Walker, Carolyn Sue.

**Project administration:** Jeremey Thomas Horne, Carolyn Sue.

**Visualization:** Jeremey Thomas Horne.

**Writing – original draft:** Jeremey Thomas Horne.

**Writing – review & editing:** Jeremey Thomas Horne, Natalie E. Allen, Serene S. Paul, Judith Walker, Carolyn Sue.

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
