## [Decision Letter · Decision Letter 0]

26 Sep 2025

PDIG-D-25-00320MITO-VATION – Feasibility of a technology-supported structured home exercise program in Mitochondrial Disease?PLOS Digital Health Dear Dr. Horne, Thank you for submitting your manuscript to PLOS Digital Health. After careful consideration, we feel that it has merit but does not fully meet PLOS Digital Health's publication criteria as it currently stands. Therefore, we invite you to submit a revised version of the manuscript that addresses the points raised during the review process. Please submit your revised manuscript within 30 days Oct 26 2025 11:59PM. If you will need more time than this to complete your revisions, please reply to this message or contact the journal office at digitalhealth@plos.org.  Please include the following items when submitting your revised manuscript: * A rebuttal letter that responds to each point raised by the editor and reviewer(s). You should upload this letter as a separate file labeled '?>Response to ReviewersRevised Manuscript with Track ChangesManuscript**Journal Requirements:**

i. State the initials, alongside each funding source, of each author to receive each grant.

ii. State what role the funders took in the study. If the funders had no role in your study, please state: “The funders had no role in study design, data collection and analysis, decision to publish, or preparation of the manuscript.”

2. Please send a completed 'Competing Interests' statement, including any COIs declared by your co-authors. If you have no competing interests to declare, please state "The authors have declared that no competing interests exist". Otherwise please declare all competing interests beginning with the statement "I have read the journal's policy and the authors of this manuscript have the following competing interests:"

3. Please ensure that your Ethics Statement is available in its entirety at the beginning of your Methods section, under a subheading 'Ethics Statement'. It must include:

1) The name(s) of the Institutional Review Board(s) or Ethics Committee(s)

2) The approval number(s), or a statement that approval was granted by the named board(s)

3) (for human participants/donors) - A statement that formal consent was obtained (must state whether verbal/written) OR the reason consent was not obtained (e.g. anonymity).

4. For studies involving third-party data, we encourage authors to share any data specific to their analyses that they can legally distribute. PLOS recognizes, however, that authors may be using third-party data they do not have the rights to share. When third-party data cannot be publicly shared, authors must provide all information necessary for interested researchers to apply to gain access to the data. (https://journals.plos.org/plosone/s/data-availability#loc-acceptable-data-access-restrictions

5. Some material included in your submission may be copyrighted. According to PLOS’s copyright policy, authors who use figures or other material (e.g., graphics, clipart, maps) from another author or copyright holder must demonstrate or obtain permission to publish this material under the Creative Commons Attribution 4.0 International (CC BY 4.0) License used by PLOS journals. Please closely review the details of PLOS’s copyright requirements here: PLOS Licenses and Copyright. If you need to request permissions from a copyright holder, you may use PLOS's Copyright Content Permission form.

Potential Copyright Issues:

a. We do not publish any copyright or trademark symbols that usually accompany proprietary names, eg (R), (C), or TM (e.g. next to drug or reagent names). Therefore please remove all instances of trademark/copyright symbols throughout the text, including Garmin®, PhysiTrack®, PhysiApp®, and Garmin Connect®.

b. Supplement 1 and 2 includes an image of an identifiable person. Please provide written confirmation or release forms, signed by the subject(s) (or their parent/legally authorized guardian), giving permission to be photographed and to have their images published under our CC-BY 4.0 license.

Otherwise, we kindly request that you remove the photograph.

**Additional Editor Comments (if provided):**

Reviewer #2:

**Reviewers' Comments:**

**Comments to the Author**

1. Does this manuscript meet PLOS Digital Health’s publication criteria?

Reviewer #1: Yes

Reviewer #2: Yes

Reviewer #3: Yes

2. Has the statistical analysis been performed appropriately and rigorously?

Reviewer #1: Yes

Reviewer #2: Yes

Reviewer #3: Yes

3. Have the authors made all data underlying the findings in their manuscript fully available (please refer to the Data Availability Statement at the start of the manuscript PDF file)?

Reviewer #1: Yes

Reviewer #2: Yes

Reviewer #3: Yes

4. Is the manuscript presented in an intelligible fashion and written in standard English?

Reviewer #1: Yes

Reviewer #2: Yes

Reviewer #3: Yes

Reviewer #1: - in baseline assessment of participants, please mention which medication did they received; for example, did they receive any AED, corticosteroids, vitamins, Coenzyme q10, L-carnitine, anti-arrhythmic drugs?

- Were there any modifications about the dietary regiment of participants during the trial period? Or did they ever received?

- Please mention the gender of participants

- Since this study measures the grip of participant, its important to mention which participant right or left handed and compare the results of non-dominant hand of each participant with their dominant one.

Reviewer #2: This is a small study of exercise in primary mild to moderate mitochondrial disease. Despite the small sample size and lack of control group, the authors have carefully summarized the data and noted the limitations in this challenging clinical setting. I have the following comments.

1. I’m not totally clear how to interpret the feasibility “goals”, most of which were not met, yet the study concludes the intervention “may be feasible”. How low of adherence would mean the intervention is not feasible?

2. Compliance drops from 87% to 23% for strength sessions over time - any indication is this because of exercise intolerance, failure of engagement in the app, something else?

3. I am unclear what “monitoring” actually means in this study - e.g., how was the monitoring (line 246-248) incorporated into the weekly catch-up calls or other contacts, in terms of reviewing actual data with participants?

4. I would actually bring the self-report survey data into the main paper, it might actually provide the most learning opportunity for readers in this space interested in building better experiences.

Minor:

1. I am curious why it makes sense to benchmark PMD against general exercise recommendations. I could not find the specific title the authors cite on the Australian government website. I do see guidance differs by age. Is there any specific data addressing that exercise levels for “all adults” applies to PMD patients?

2. The figures and tables were extremely grainy making them hard to read.

3. Was it surprising that 2 of the 10 seem to have 3-4 hours of intense activity per day (Figure 3)?

4. Sounds like the subjects could choose video or phone, but the term “Telehealth” refers just to video for some reason? Line 347. Can the authors provide a sentence as to why video was not acceptable?

5. Not sure I would use drop-out rate as a measure of acceptability. e.g., if compliance were zero but the subjects simply did not take the step to actually drop out, this does not seem to argue the protocol was acceptable?

Reviewer #3: Thank you for the opportunity to review this interesting research, please read the following comments to strengthen you findings:

Introduction: to better establish the context and significance of the research, please consider adding recent statistics on the worldwide prevalence and impact of PMD.

Methods:

Lines 121-122: Please provide a clear rationale for selecting the Garmin Vivoactive 5 smartwatch and the PhysiTrack platform. It would be helpful to mention if these specific tools have been validated or used in previous research and to briefly explain why they were considered optimal for this study's objectives compared to other available options.

Lines 124, 177, 179: There appears to be a discrepancy regarding the study duration. The text states an 8-week period, but the mention of "Week 0" for baseline and "Week 9" for follow-up suggests a total duration of 10 weeks. Please clarify the exact timeline of the intervention and data collection to ensure consistency.

Analysis of Potential Confounders (Baseline and Demographics):

1- Baseline Measures: The authors should clarify if there were any significant differences in baseline measures among the participants. If so, please discuss how these differences were controlled for in the analysis or how they might have influenced the study's outcomes.

2- Influence of Age: The manuscript would be strengthened by an analysis of the potential influence of age on the study's outcomes. As a key demographic variable, it is important to determine if age has a significant effect on the results. I would recommend conducting a supplementary statistical analysis to explore this. For instance, the authors could stratify participants into two or three age-based categories and compare the outcomes using an appropriate test (e.g., t-test for two groups or ANOVA for three or more). Alternatively, grouping participants by other relevant variables—such as NMDAS score, exercise history, or clinical presentation—and examining the role of age within these groups could also yield valuable information. This analysis could contribute significantly to the findings by providing insights that may help refine or tailor the treatment program settings.

**Do you want your identity to be public for this peer review?** For information about this choice, including consent withdrawal, please see our Privacy Policy

Reviewer #1: **Yes:** Hamidreza Ashayeri

Reviewer #2: No

Reviewer #3: **Yes:** Jehad Omar Abualrob

**Figure resubmission:****Reproducibility:** To enhance the reproducibility of your results, we recommend that authors of applicable studies deposit laboratory protocols in protocols.io, where a protocol can be assigned its own identifier (DOI) such that it can be cited independently in the future. Additionally, PLOS ONE offers an option to publish peer-reviewed clinical study protocols. Read more information on sharing protocols at https://plos.org/protocols?utm_medium=editorial-email&utm_source=authorletters&utm_campaign=protocols

---

## [Editor Report · Decision Letter 1]

6 Feb 2026

MITO-VATION – Feasibility of a technology-supported structured home exercise program in Mitochondrial Disease?

PDIG-D-25-00320R1

Dear Mr Horne,

We are pleased to inform you that your manuscript 'MITO-VATION – Feasibility of a technology-supported structured home exercise program in Mitochondrial Disease?' has been provisionally accepted for publication in PLOS Digital Health.

Best regards,

Hisham Al-Obaidi, PHD

Academic Editor

PLOS Digital Health